# Impact of artificial feeding on the developmental cycle of two triatomine species

Raquel Aparecida Ferreira *, Ana Carolina Corrêa Nascimento[º],
Natalya Lorena da Silva Santos[º], Liléia Gonçalves Diotaiuti,
Eduardo Fernandes e Silva, Silvia Ermelinda Barbosa*

Grupo de Pesquisa Triatomíneos, Instituto René Rachou/Fiocruz Minas, Belo Horizonte, Minas Gerais, Brasil

[º] These authors contributed equally to this work.
* raquel.ferreira@fiocruz.br (RAF); silvia.leite@fiocruz.br (SEB)

## Abstract

### Background

Live animals are the preferred food source for rearing and keeping triatomines in insectaries. However, there is an increasing demand for artificial feeding as an alternative method to using live animals. Despite this demand, scientific studies showing the viability of rearing triatomine colonies in insectaries with artificial feeding are still lacking. In this regard, we tested whether feeding triatomines with an artificial system affects different physiological and biological parameters during the insects' development.

### Methodology/principal findings

Three experimental groups were established for the species *Triatoma infestans* and *Panstrongylus megistus*. The insects were fed weekly and individually, with each group receiving different treatments, as follows: A) artificial feeding system with rabbit blood, B) live animal host, and C) alternating feeding (one week A and the next B). The groups were kept under the same environmental conditions and monitored daily. The parameters evaluated per instar and total cycle were the effect of the blood supply method on the insects' developmental time, the association between feeding method and insect mortality, weight gain, and the number of feedings needed to trigger molting. The latter two were evaluated only in *P. megistus*.

### Conclusions/significance

Sex did not influence the insects' survival or weight gain. The biological cycle duration was shortest in insects fed on a live animal host, followed by the group fed alternately, and longest in those fed artificially. *Panstrongylus megistus* fed on an artificial system required more feedings to complete molting and gained less weight. The

**Data availability statement:** All relevant data are within the paper and its Supporting Information files.

**Funding:** Financial support of Fundação de Amparo a Pesquisa de Minas Gerais (FAPEMIG), Instituto René Rachou (IRR) and Coordination of health surveillance and reference laboratories (Fiocruz).

**Competing interests:** NO authors have competing interests.

insect mortality rate was not influenced by the feeding method. This is the first study showing that artificial feeding conditions affect different physiological parameters in two triatomine species, questioning the viability of exclusive artificial feeding as a sustainable strategy in a laboratory environment.

## Introduction

Triatomines (Reduviidae: Triatominae Jeannel, 1919) are hemimetabolous insects that exhibit five nymphal stages in addition to the adult stage. Both sexes are obligatory blood-feeders throughout all life stages, feeding in their natural environment on a wide range of terrestrial and arboreal mammals (especially didelphids, edentates, and rodents), birds, and bats [1]. In domestic settings, triatomines feed on domestic animals and humans [1–2], posing a significant public health concern as vectors of trypanosomatids, including *Trypanosoma cruzi* Chagas, 1909, the etiological agent of Chagas disease [3], one of the most important parasitic diseases in Latin America [4].

The subfamily Triatominae comprises around 157 described species, grouped into 18 genera, with 66 species occurring in Brazil [5–8]. Although all triatomine species are considered vectors of *T. cruzi*, their vector competence varies depending on biological and behavioral factors.

In general, scientific literature shows that the feeding performance of triatomines differs between species due to factors such as size and nymphal stage, type of blood, mating status, host physiology, insect's feeding apparatus, and ambient temperature, among other factors [9–14]. Therefore, a fundamental requirement for successfully rearing these insects in insectaries is a thorough understanding of their biology [15].

Rearing triatomine colonies in insectaries is a labor-intensive task. While some species that are better adapted to artificial conditions (temperature and humidity) can be managed relatively easily, others pose significant challenges, as they require not only environmental conditions similar to their natural habitat but also specific food sources. In this regard, not all triatomine species adapt well to insectary conditions, and specific modifications may be necessary to meet the unique requirements of each species [16]. Although there is extensive literature on the biology of triatomines [17–24], little is known about the biology of species fed artificially—using artificial apparatus rather than live animal hosts—over several generations and reared in insectaries. This knowledge gap largely stems from the varying rearing methodologies, which complicate data comparison [25–26].

In addition to the effort required to rear triatomines, it is also necessary to maintain the blood source animals. While there is a perception, not yet fully documented in the scientific literature, that triatomines develop better when fed on live animals, this method is not always considered viable or ethical. In response to these concerns, ethical committees are increasingly demanding alternative feeding methodologies to replace the use of live animals.

Ideally, physiological and biological parameters related to the developmental cycle of insects should remain unaffected by alternative feeding methods, such as artificial

feeders, facilitating the routine replacement of live animals in insectaries. In this context, we tested the hypothesis that feeding triatomines with an artificial system, rather than a live animal host, would not impact physiological and biological parameters related to the developmental cycle of two triatomine species: *Panstrongylus megistus* Burmeister, 1835, and *Triatoma infestans* (Klug, 1834). *Panstrongylus megistus* is a native species widely distributed across Brazil and commonly associated with high infection rates of *T. cruzi*. It is currently the primary species of epidemiological importance in the country [27]. In contrast, *T. infestans*, although exotic, was the most significant species until the 2000s, responsible for about 80% of Chagas disease cases in Brazil during the 1970s and 1980s [28]. While *T. infestans* is now virtually controlled in Brazil, it continues to be the primary species of epidemiological importance in various Latin American countries [29–30].

Despite the existing knowledge about the biological cycle of some triatomine species fed artificially in the laboratory [23,31–33], gaps remain in the scientific literature concerning specific physiological parameters of insects raised through artificial methods that need to be better documented, especially because previous studies have not analyzed some physiological parameters together. In this scenario, we aimed to compare the biological aspects of the developmental cycle of *P. megistus* and *T. infestans* when fed on live animal hosts versus an artificial feeder. Here, an in-depth study encompassing and analyzing several critical physiological development parameters in triatomines is conducted using robust mathematical models and analyses. Through the findings, it is possible to predict the long-term impact of artificial feeding on the biological cycle of two of the most important triatomine species in the context of Chagas disease transmission in Brazil.

## Materials and methods

### Ethical aspects

All experiments involving live animals were conducted in accordance with animal experimentation guidelines and were approved by the Ethics Committee on the Use of Laboratory Animals (CEUA/FIOCRUZ) under protocol number LW 10/18.

### Insectary description

The Triatomine Insectary at the René Rachou Institute (IRR) currently rears 30 species of triatomines, meeting the research demands of various institutional research groups, training and qualification of healthcare professionals within Brazil's Unified Health System (*Sistema Único de Saúde*—SUS), and the enhancement of specimens for the Coleção de Vetores de Tripanosomatídeos [34–35].

In this facility, semi-controlled conditions are maintained with a temperature range of 26–28°C and relative humidity of 60–70%, which align with the optimal levels known for most species [36]. The blood sources commonly used in the laboratories and our insectary are pigeons (*Columba livia*) and chickens (*Gallus gallus*) [37–38]. These hosts are selected for their ease of breeding in the animal facility and their resistance to *T. cruzi*.

### Experimental model

*Panstrongylus megistus* and *T. infestans* have been reared in the IRR insectary for decades, adapting well to the bioclimatic conditions of the facility. These species have been maintained for multiple generations and are used as experimental models for various scientific research studies conducted at the institution.

Newly laid eggs (n = 200) from these species were randomly collected from the insectary colonies. The eggs were observed daily until hatching, and the first instar nymphs that emerged on the same day were individually placed in acrylic pots (4x2.5 cm). Each pot contained filter paper at the bottom to retain moisture from the insect's urine and feces, and a piece of folded cardboard was included to allow the insect to move up to the lid. The lid was made of cotton fabric to permit the insect's mouthparts to pass through for feeding.

## Experimental conditions

Three groups of 50 triatomines of each species were established and fed using different methods: Group A received blood from anesthetized chickens (ketamine + detomidine), Group B was fed using an artificial feeder with citrated rabbit (*Oryctolagus cuniculus*) blood, and Group C was fed alternately—one week on chickens and the next week on the artificial feeder. Each pot was labeled with an individual code corresponding to the experimental group to facilitate tracking in a spreadsheet. Insects remained in their assigned group and pot until reaching the adult stage.

The first feeding for the insects of each instar occurred five days after egg hatching, with Groups A and C feeding on chickens and Group B on the artificial feeder. If insects did not feed on this initial day, they were offered food daily until they fed. Following this initial feeding, subsequent meals were provided weekly, with each insect allowed to feed for 30 minutes. During the first feeding, the insects were allowed to consume blood to their full capacity.

All groups of insects were maintained under semi-controlled conditions with a temperature of 27 ± 2°C and relative humidity of 50 ± 20%. Insects belonging to the same species were fed on the same days. Daily checks were conducted to record occurrences of molting and deaths. An insect was considered dead if it did not respond to external stimuli and remained immobile on the pot or the surface of the filter paper.

Although the sex of triatomines can be determined starting from the first instar, we opted to sex them only at the adult stage to minimize excessive handling of the specimens.

The blood used in the artificial feedings was 10% citrated rabbit blood (25g P.A. citrate, 8g citric acid, and 24.5g glucose). This blood is collected, prepared, and sterilized at the Institute of Science and Technology in Biomodels (ICTB)/Fiocruz. It is packaged in an autoclavable glass bottle with a rubber cap, sealed, and distributed chilled on dry ice to the laboratories. In the laboratory, the blood is immediately stored at -20°C and used within 10 days of being collected.

The artificial feeding device used consisted of a glass apparatus, made by a craftsman in the shape of a baby bottle, with an external compartment for circulating water heated by an attached water bath and an internal compartment containing the blood. The water temperature was set at 40°C, and the blood temperature in the bottles remained at 39.5°C. The membrane used to cover the feeding bottle was a latex membrane. These were obtained from natural latex gloves, which were cut into 10x10 cm pieces and washed with Type I water to remove dust. The dried latex membrane was used to seal the mouth of the glass apparatus with the help of an elastic band, simulating the skin of a live animal that was pierced by the insect's mouthparts during feeding [15].

The following physiological parameters were evaluated for both the triatomine species at each developmental stage: the effect of different blood offerings (treatments A, B, or C) on the insects' developmental time, the association between treatments and insect mortality rate, and survival time between sexes. Exclusively for *P. megistus*. the number of feedings required to trigger molting was recorded for each blood treatment. Additionally, weight changes before and after feeding were measured, and weight gain was analyzed in relation to sex. The specimens were weighed on a precision scale (AGZN200) before and after feeding.

## Statistical analyses

In this study, Generalized Estimating Equations (GEE) modeling, Cox modeling, and parametric model analyses were processed using R software (version 4.2.1). Graphs were created with Prisma (version 8.02), and survival curve statistical analyses were conducted using SPSS software (version 18.5). Regardless of the model or test used, a significance level of 5% was adopted in all statistical analyses performed.

## Effect of different blood offerings on the developmental time of *T. infestans* and *P. megistus*

For each instar and species, the Kaplan-Meier estimator was used to illustrate the percentage of insect survival days across the three treatments (A, B, and C). This analysis was also applied to the entire life cycle of each species, from

the first instar nymph to the adult stage molting. Mortality data at each instar were censored at that stage and excluded from subsequent analyses. For the total lifespan of the insects (from all instars until molting to the adult stage), data were compiled from all individuals, summing the days of life from the first instar to either the point of death or reaching the adult stage.

The Log-rank (Mantel-Cox), Gehan-Breslow-Wilcoxon, and Tarone-Ware tests were used to compare the development curves across the three treatments. The Log-rank test is a hypothesis test for comparing the survival distributions (considered here as the developmental time of insects) of two or more treatments, assigning equal weight to the event (instar molting) at all time points. The Gehan-Breslow-Wilcoxon test gives more weight to events (instar molting) that occur at earlier time points and does not assume a consistent hazard ratio; instead, it requires that one group has a higher risk than the other. The Tarone-Ware test is more sensitive to differences at the end of the curve related to the event.

### Survival time according to sex in *T. infestans* and *P. megistus*

This analysis aimed to compare survival time between sexes, defined as the time until the insect reaches adulthood. Direct survival measurements could not be used because one of the basic assumptions of this methodology is that the covariates must be measured at baseline. However, determining the sex of the insects at the nymphal stage is not possible. Therefore, the gamma regression model was adjusted for this evaluation, and the non-parametric Wilcoxon test was performed. In these analyses, the variable 'time until the insect becomes an adult' was entered into the model on a logarithmic scale. However, the interpretation should be based on the original scale, as shown in the results.

### Association between treatments and mortality rate in *T. infestans* and *P. megistus*

To evaluate the effect of different blood presentation methods on the insect mortality rates (group-by-group comparison), the Cox model was used if the assumption of proportional hazards was met, as assessed by the Schoenfeld residual test. In the absence of proportional hazards, parametric models were employed as alternatives. In this latter case, Weibull, Log-normal, and exponential distributions were tested, and the best-fitting models were selected. Unlike the Cox model, these parametric models assume a specific probability distribution for survival time, with interpretations based on median times. To assess the adequacy of the Weibull model, the survival of Cox-Snell residuals estimated by the Kaplan-Meier method was compared to the estimates obtained from the Weibull model.

The interpretation of the Cox model relies on the hazard ratio, which is always a positive number. When this ratio was less than 1, we inverted the value of the baseline category for interpretative purposes to better illustrate the effect size.

### Comparison of the number of feedings required for molting in *P. megistus* across the three treatments

The GEE approach, the most suitable modeling method for analyzing longitudinal data with repeated measurements of the same samples, was adopted to evaluate the number of feedings required for molting in *P. megistus* across the three treatments. Previous measurements of insects that died during the experiment were included in our analyses. A detailed description of the test can be found in the supplementary material (S1Text).

A comprehensive range of GEE models assuming different distributions and covariance matrices was tested for each variable. The best-fitting model was selected based on the lowest quasi-likelihood under the independence model criterion (QIC). In this analysis, Poisson regression via GEE was used, with the outcome (data used in the analyses) being the count of feedings needed for an insect to molt from one instar to the next. The study was unbalanced due to insect mortality over time. The artificial feeder group was used as the reference category for the experimental group, based on the hypothesis that this feeding method affects insect development, and the first instar (N1) served as the reference stage.

In the model interpretation, we considered the interaction between the instar stage and the feeding group. Due to significant interactions, differences between the groups could not be interpreted without accounting for each instar stage. Consequently,

the analysis involved comparing group A vs. B and group B vs. C, with results adjusted by adding the estimated values from these comparisons to those obtained from the interactions specific to each instar stage and feeding group.

### Weight gain in different instars of *P. megistus*

The GEE model was adjusted using a normal distribution and the identity link function for all analyses, including each instar and the entire developmental cycle. The model was run twice: in the first run, the artificial feeder group served as the reference category, and N1 was used as the reference for the instar. In the second run, while N1 was kept as the reference for the instar, the reference category for the experimental group was changed to the alternately fed group. For model interpretation, the estimated values from comparisons between group A vs. B or group B vs. C were used, and these were combined with the estimates obtained from interactions specific to each instar and group.

### Weight gain based on sex in *P. megistus*

The sex of the triatomines was determined only at the adult stage. Therefore, the analysis of the influence of sex on weight gain considered the total weight gain throughout the insect's entire developmental cycle. Since each individual contributed only once to the sample and the data lacked a longitudinal structure, the method of blood presentation was not included in this analysis.

For this dataset, none of the regression model assumptions were met; therefore, the non-parametric Wilcoxon test was used.

## Results

### Effect of different blood offerings on insect developmental time

We observed that 69 specimens of *T. infestans* and 49 of *P. megistus* reached the adult stage. For the three types of blood offerings, the average developmental time from N1 to the adult stage was estimated at 1920 days for *T. infestans* and 1310 days for *P. megistus* (S1 Table).

In *T. infestans*, the development of nearly all instars (N2, N3, N4, and N5), as well as the overall developmental cycle from N1 to adulthood, was temporally affected by the method of blood offering (Fig 1 and Fig 2). Development was fastest when blood was provided by a live animal host, followed by alternating feeding, and slowest when blood was offered exclusively through an artificial feeder (statistical tests presented in figure legends). For N1, we found no significant differences in the developmental time among the experimental groups throughout the period (Fig 1A). However, when examining the timing of molts at both the beginning and end of the observation period, faster development was observed when blood was provided by a live animal host, followed by alternating feeding, and slowest with artificial feeding (see figure captions for statistical details).

For *P. megistus*, we observed that the development of almost all instars (N1, N2, N3, and N4) and the overall developmental cycle was temporally affected by the method of blood offering (Figs 3 and 4). Development was fastest when blood was provided by a live animal host, followed by alternating feeding, and slowest when blood was offered exclusively through an artificial feeder. Statistical tests are detailed in the figure legends. For N5, there was no significant difference in developmental time between the live animal host and alternating feeding groups; however, a significant difference was found when compared to the artificial feeder group (Fig 3E).

### Survival rate according to sex

Sixty-nine specimens of *T. infestans* reached the adult stage, with 35 from the chicken-fed group, 25 from the alternately-fed group, and nine from the artificial feeder group. No significant difference in survival rate was observed between the sexes (estimate = 0.8946, p = 0.2556, CI = [0.73961; 1.08214]).

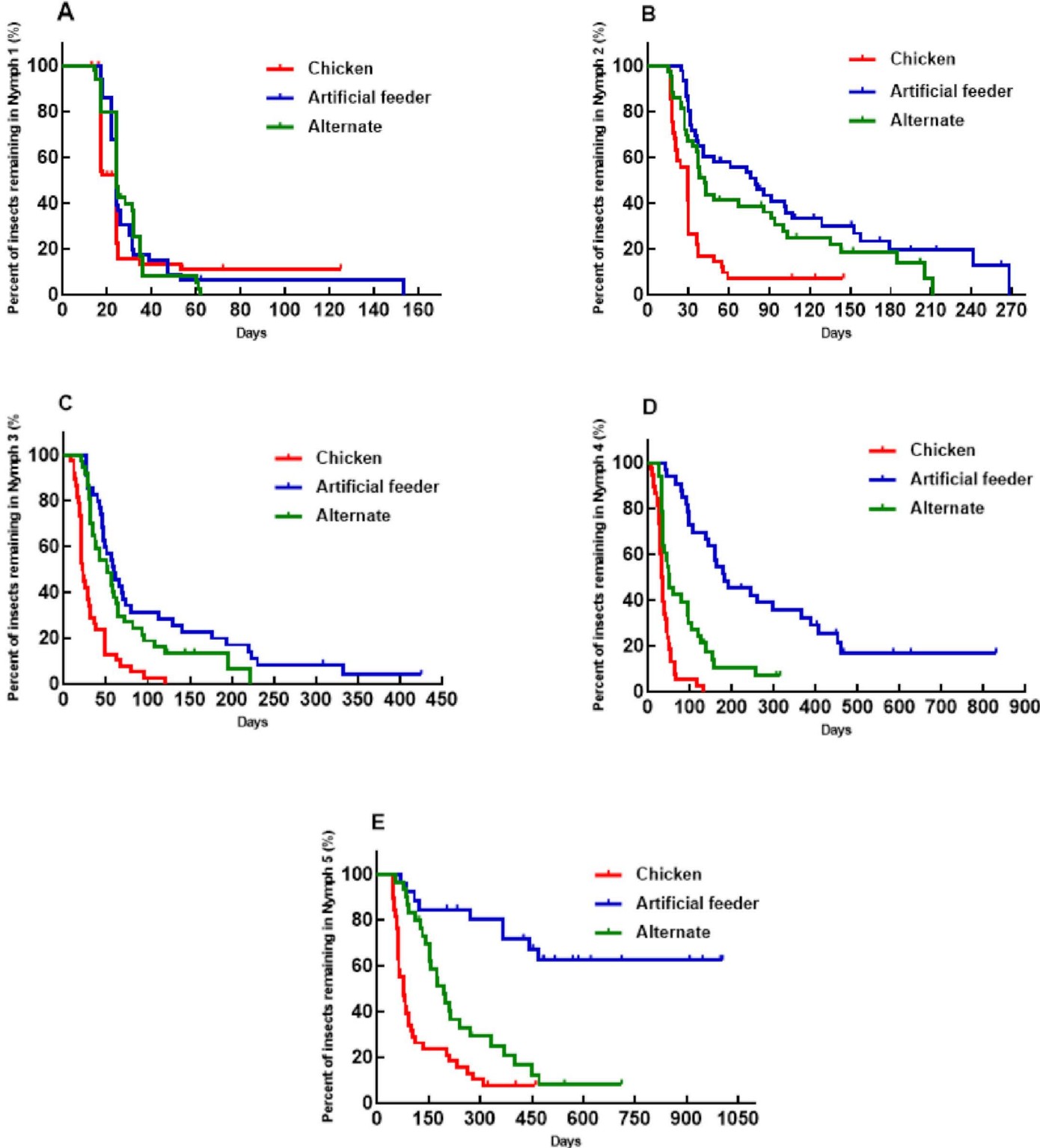

**Fig 1. Percentage of *Triatoma infestans* individuals remaining in each instar over time (in days).** The red line represents insects fed on chickens, the blue line represents insects fed using an artificial feeder, and the green line represents insects fed alternately (chickens or artificial feeder). Dashes indicate censored events. **A)** Time in N1 until molting to N2 (Log-rank test, p = 0.386; Breslow test, p = 0.006; Tarone-Ware test, p = 0.031). **B)** Time in

N2 until molting to N3. **C)** Time in N3 until molting to N4. **D)** Time in N4 until molting to N5. **E)** Time in N5 until adulthood. The Log-rank, Breslow, and Tarone-Ware tests showed significant differences with p-values less than 0.0001 for all tests conducted, except for panel **A.**

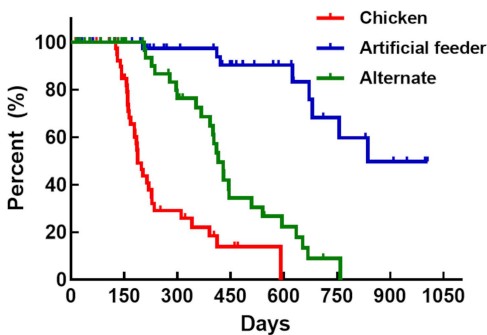

**Fig 2. Developmental time of *Triatoma infestans* from N1 to adulthood.** The red line represents insects fed on chickens, the blue line represents insects fed using an artificial feeder, and the green line represents insects fed alternately (chickens or artificial feeder). Dashes indicate censored events (p<0.0001 for Log-rank, Breslow, and Tarone-Ware tests).

Forty-nine specimens of *P. megistus* reached adulthood. No significant difference in survival rate was observed between the sexes (estimate = 0.9840, p = 0.8894, CI = [0.78542; 1.23288]).

The absolute number of insects that survived according to sex for the studied species is showed in S1 Fig, S2 Fig.

## Association between treatments and insect mortality rate

Table 1 presents the values related to the comparison of the death rates of insects fed on chicken or alternately on chicken and an artificial medium, for the two species of triatomines.

Figs 5 and 6 show the absolute numbers of deaths in all instars and the entire developmental cycle according to treatment. Figs 7 and 8 show the total number of deaths across all instars and throughout the developmental cycle, according to the treatment in *P. megistus*.

## Comparison of the number of feedings required to trigger molting in insects across the three treatments

Significant interactions were found between the developmental instars and groups (S2 Table). Thus, data interpretation considered the interaction between the groups (different blood presentation methods) and the stages of the insect's developmental cycle. Specifically, the estimated values for comparisons between the chicken group and the artificial feeder group or the alternating group and the artificial feeder group (S2 Table) were combined with estimates from the interactions of specific developmental stages and groups (S2 Table). Table 2 describes the values obtained from this interpretation, indicating the number of additional feedings required by insects fed under artificial conditions to trigger molting.

## Weight gain in different instars of *P. megistus*

S3 Table presents the descriptive measures of weight gain across the different instars of *P. megistus*. At the end of the N1 instar, 142 insects remained alive and were included in the analyses, with each individual receiving a maximum of 6 feedings. In N2, 92 insects were analyzed, each with up to 12 feedings. In N3, 76 insects were included, with a maximum of 18 feedings per individual. Finally, in N4 and N5, 67 and 61 insects, respectively, survived until the end of the instar, with the maximum number of feedings per individual being 16 in N4 and 18 in N5.

Fig 9 illustrates the weight gain in each instar of *P. megistus* based on the type of blood feeding received.

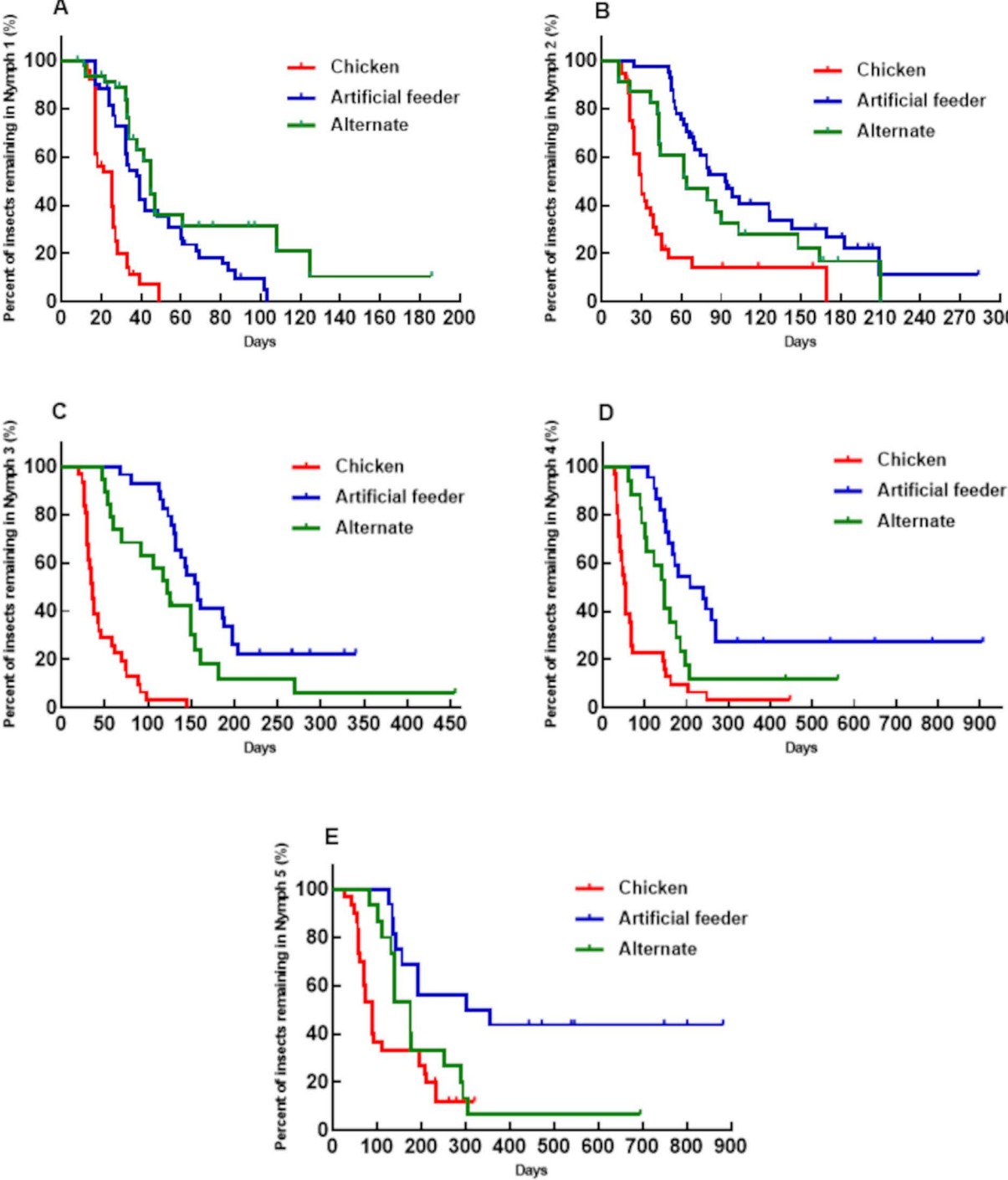

**Fig 3. Percentage of *Panstrongylus megistus* individuals remaining in each instar over time (days).** The red line represents insects fed on chickens, the blue line represents insects fed using an artificial feeder, and the green line represents insects fed alternately (chickens or artificial feeder). Dashes indicate censored events (deaths). **A)** Time in N1 until molting to N2. **B)** Time in N2 until molting to N3. **C)** Time in N3 until molting to N4. **D)** Time in N4 until molting to N5. **E)** Time in N5 until reaching adulthood (Log-rank, Breslow, and Tarone-Ware tests, p < 0.0001 for all tests conducted, except for N5. For N5, Breslow p = 0.0006 for the artificial feeder vs. chicken group comparison, and Log-rank p = 0.014 for the artificial feeder vs. alternating group comparison. The other comparisons were not significant).

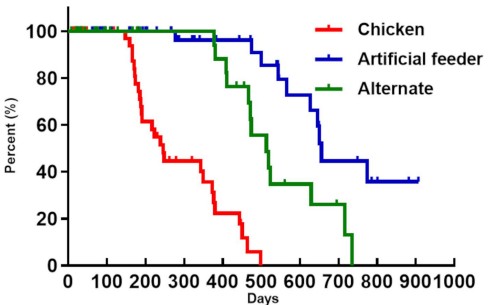

**Fig 4. Developmental time of *Panstrongylus megistus* from N1 to adulthood.** The red line represents insects fed on chickens, the blue line represents insects fed using an artificial feeder, and the green line represents insects fed alternately (chickens or artificial feeder). Dashes indicate censored events (deaths) (Log-rank, Breslow, and Tarone-Ware tests, p<0.0001).

**Table 1. Comparison of the mortality rate of insects fed on chickens or alternately with the mortality rate of insects fed using an artificial feeder (reference group for analysis).**

| Species | Instar | Comparison (type of blood offering) | Mortality rate | p-value | CI (95%) |
|---|---|---|---|---|---|
| *Triatoma infestans* | N1 | artificial x alternated | 1.9868 | 0.2793 | [0.57286; 6.89094] |
| | | artificial x chicken | 2.5886 | 0.1183 | [0.78472; 8.53931] |
| | N3 | artificial x alternated | 14.1344 | **0.0289** | [1.27381; 1937.643] |
| | | artificial x chicken | 6.8831 | 0.3690 | [0.03687; 1284.565] |
| | N4 | artificial x alternated | 2.1392 | 0.3153 | [0.45484; 8.80895] |
| | | artificial x chicken | 0.7358 | 0.8427 | [0.00515; 9.71894 |
| | N5 | artificial x alternated | 0.3473 | 0.1029 | [0.09743; 1.23798] |
| | | artificial x chicken | 1.0366 | 0.9592 | [0.26184; 4.10381] |
| | N1–adult | artificial x alternated | 1.0738 | 0.8345 | [0.55073; 2.09358] |
| | | artificial x chicken | 1.0715 | 0.8354 | [0.55863; 2.05511] |
| *Panstrongylus megistus* | N1 | artificial x alternated | 2.8643 | **0.0068** | [1.33749; 6.1339] |
| | | artificial x chicken | 3.8576 | **0.0029** | [1.58624; 9.38117] |
| | N2 | artificial x alternated | 0.9175 | 0.8818 | [0.29502; 2.85343] |
| | | artificial x chicken | 2.9254 | **0.0413** | [1.04314; 8.20406] |
| | N3 | artificial x alternated | 0.5347 | 0.3751 | [0.09877; 2.03414] |
| | | artificial x chicken | 0.4368 | 0.5353 | [0.00329; 4.01933] |
| | N4 | artificial x alternated | 1.6799 | 0.5464 | [0.27256; 8.88465] |
| | | artificial x chicken | 1.6285 | 0.6382 | [0.15257; 10.76077] |
| | N5 | artificial x alternated | 0.2123 | **0.0459** | [0.02234; 0.975] |
| | | artificial x chicken | 1.0255 | 0.9693 | [0.26819; 3.76491] |
| | N1–adult* | artificial x alternated | 0,4760 | **0.0376** | [0.23644; 0.9582] |
| | | artificial x chicken | 0.7837 | 0.5125 | [0.37793; 1.62519] |

Cox model. *Log-normal parametric model. CI = confidence interval. **Bold**: significant difference.

From the models run, several significant interactions were observed between the experimental groups and the different nymphal instars in *P. megistus* (S4 Table). In interpreting the model, the estimated value obtained from the comparison between groups A x B or B x C (S4 Table) was added to the estimate obtained from the interaction of that specific stage and group (S4 Table). Table 3 shows the values obtained from this interpretation.

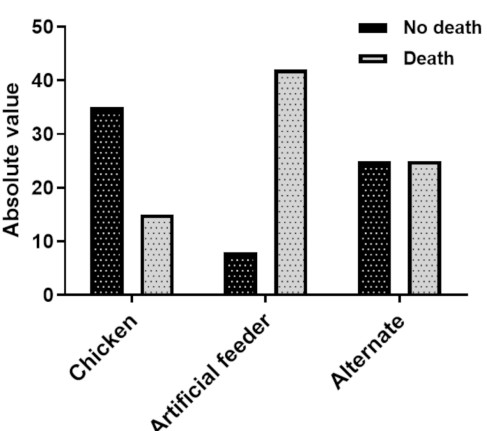

**Fig 5. Number of *Triatoma infestans* individuals that died or survived according to treatment (type of blood offering). A)** N1; **B)** N2; **C)** N3; **D)** N4; **E)** N5.

**Fig 6. Number of *Triatoma infestans* individuals that died or survived throughout the developmental cycle (N1–adult) according to treatment.**

Fig 7. **Number of *Panstrongylus megistus* individuals that died or survived according to treatment (type of blood offering). A)** N1; **B)** N2; **C)** N3; **D)** N4; **E)** N5.

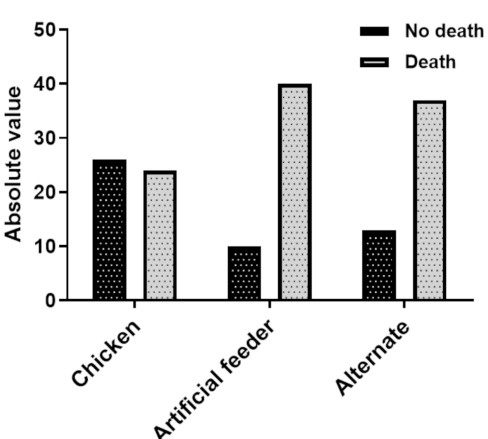

Fig 8. **Number of *Panstrongylus megistus* individuals that died or survived throughout the developmental cycle (N1–adult) according to treatment.**

**Table 2. Number of additional feedings required by insects fed under artificial conditions to trigger molting throughout the developmental cycle of _P. megistus_.**

| Description of Interaction | Number | p-value | IC* (95%) |
| --- | --- | --- | --- |
| N1: Chicken | 2.49 | < 0.001 | [2.10; 2.94] |
| N1: Alternated | 1.24 | 0.04 | [0.1; 1.52] |
| N2: Chicken | 2.72 | < 0.001 | [2.13; 3.47] |
| N2: Alternated | 1.47 | < 0.001 | [1.16; 1.86] |
| N3: Chicken | 3.62 | <0.001 | [2.94; 4.47] |
| N3: Alternated | 1.87 | < 0.001 | [1.51; 2.31] |
| N4: Chicken | 2.62 | < 0.001 | [2.08; 3.29] |
| N4: Alternated | 1.64 | < 0.001 | [1.27; 2.12] |
| N5: Chicken | 1.40 | 0.027 | [1.4; -1.91] |
| N5: Alternated | – | n.s. | – |

GEE Poisson model interpretation; *CI: confidence interval. n.s.: not significant.

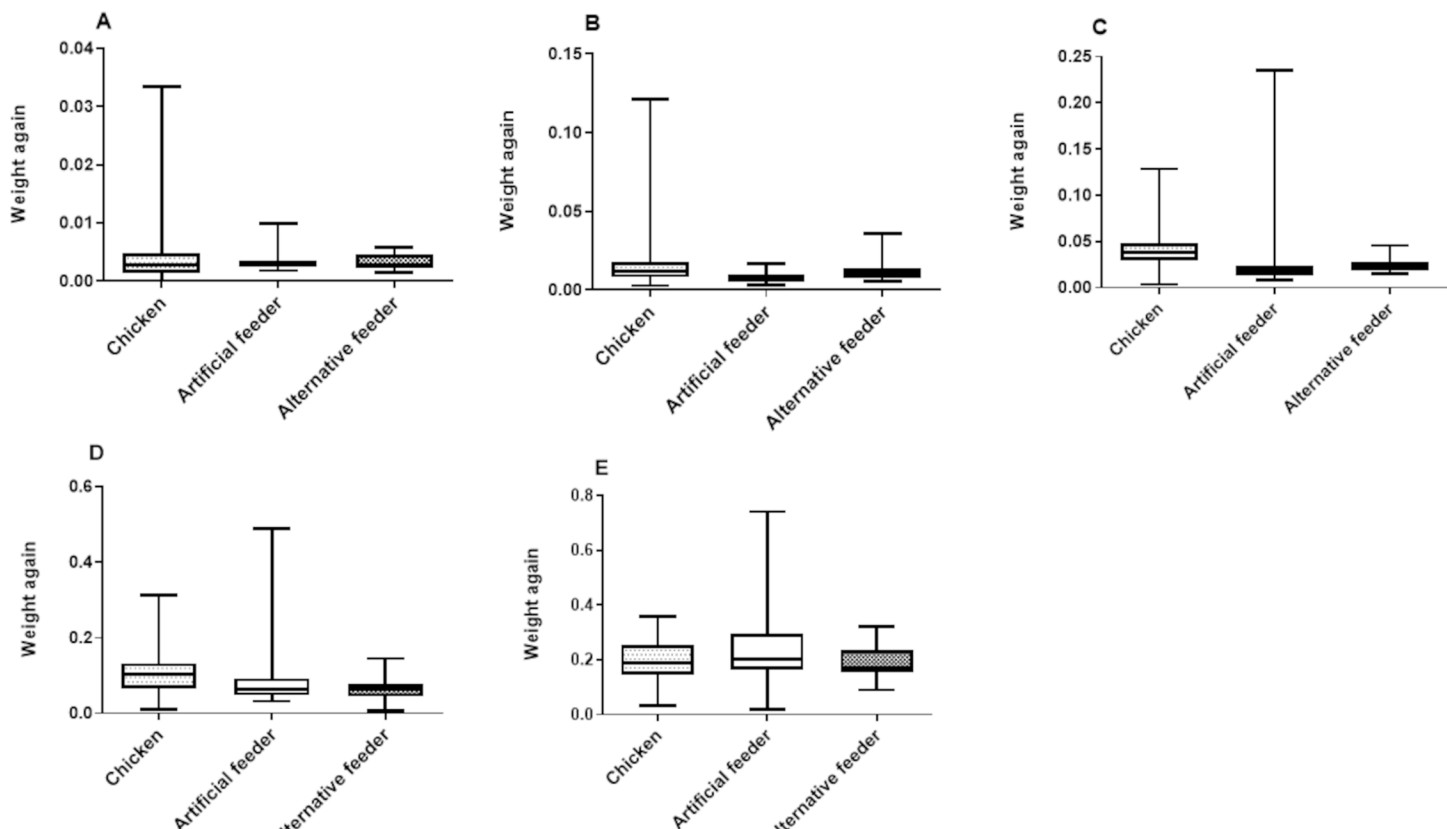

**Fig 9. Box plots showing weight gain across different experimental groups and instars in _P. megistus_.** The weight gain is depicted for the following instars: **A)** N1, **B)** N2, **C)** N3, **D)** N4, **E)** N5. The graphs present the median of the data (indicated by the horizontal line within the bars). The portion below the median represents the first quartile, while the portion above the median represents the third quartile. The minimum and maximum values of the data are represented by the vertical lines extending from the bars, delineating these values on the graph.

**Table 3. Interpretation of weight gain of N1–N5 nymphs in *P. megistus*. The reference instar in all analyses was N1.**

| Type of Blood Offering Fixed | Group Interaction | Mean Weight Gain (units) | p-value | IC* (95%) |
|---|---|---|---|---|
| Artificial feeder | Alternated | – | n.s. | _ |
| | Chicken | -- | n.s. | _ |
| | N2 | 0.004 | 0.001 | [0,002; 0.007] |
| | N3 | 1.47 | < 0.001 | [1.16; 1.86] |
| | N4 | 0.021 | <0.001 | [0,010; 0,033] |
| | N5 | 1.87 | < 0.001 | [1,51; 2,31] |
| | Alternated: N2 | 0,034 | < 0.001 | [0,015; 0,0541] |
| | Chicken: N2 | 1.64 | < 0.001 | [1,27; 2,12] |
| | Alternated: N3 | – | 0.524 | [IC= -0,071; 0,036 e p= 0,524] |
| | Chicken: N3 | – | n.s. | – |
| Alternated | Alternated: N4 | – | 0.562 | -0.00325; 0.00598] |
| | Chicken: N4 | – | < 0.001 | [0.0056; 0.0174] |
| | Alternated: N5 | 0.0184 | 0.002 | [0.00653; 0.0303] |
| | Chicken: N5 | 0.026 | 0.016 | [0.0048; 0.0483] |
| | Chicken | – | 0.818 | [-0.0344; 0.0435] |

GEE Poisson model interpretation; *CI: confidence interval. n.s.: not significant.

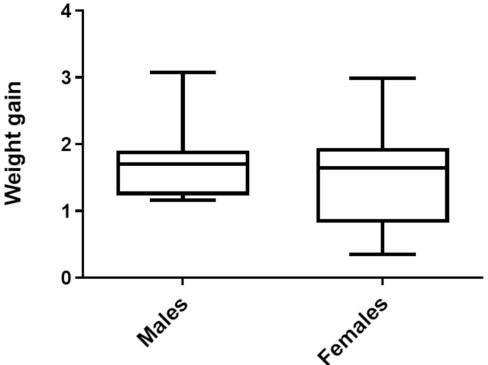

**Fig 10. Box plots showing weight gain in *P. megistus* according to sex.** The graphs display the median weight gain (horizontal line within the bars). The portion below the median represents the first quartile, while the portion above the median represents the third quartile. The minimum and maximum values of the data are represented by the vertical lines extending from the bars, delineating these values on the graph.

## Influence of sex on weight gain

Fig 10 illustrates the weight gain distribution of *P. megistus* according to sex, regardless of the feeding method. The weight gain data for the 38 adult insects, including 16 males and 22 females, is shown in S5 Table.

There was no significant difference in the weight gain of insects throughout their lives based on sex (p = 0.438) (Fig 10).

## Discussion

Historically, the use of animals for scientific experimentation has always been the subject of heated debates worldwide [39–42]. There is a consensus that the experimental use of animals has enabled and facilitated advances in several fields of science, and continues to drive discoveries, particularly in medical and biological sciences. In contrast, several

ethical, legal, moral, and even philosophical aspects permeate the discussions, raising the question of the need to replace animal use with alternative resources or strategies [39–42]. In line with these issues, brazilian legislation is increasingly demanding the replacement of live animals used to feed triatomines with artificial feeding methods. This shift aligns with the national and international laws based on the universal doctrine proposed by Russell and Burch (1959) [43] known as principle of the 3 Rs (Replacement, Reduction, and Refinement). The Brazilian 'Arouca Law' is a milestone in legislation regarding the use of animals in research, as it established the 'National Council for the Control of Animal Experimentation' (CONCEA), which addresses legal and ethical issues related to the use of experimental animals in scientific research and other related activities [44]. The Brazilian scientific community is committed to adhering to these ethical standards, demonstrating a strong dedication to the responsible conduct of research. In fact, this commitment permeates the entire global scientific community, which is increasingly focused on replacing or reducing the use of animals with techniques such as human cell lines, the use of cadavers for studies, computer simulations, mathematical models, and nanotechnology, for example [44–45].

From this perspective, reducing the reliance on live animals as blood sources, refining their use, and exploring suitable replacements is essential. This study tested these efforts for triatomines. Our findings demonstrate that the species studied, despite belonging to different genera, exhibited similar responses to the type of blood offering.

We observed significant differences between the experimental groups in both species across all evaluated parameters: developmental time, the number of feedings required to trigger molting, and weight gain. The only parameter with inconsistent and minimal differences between the species was the survival rate of the insects.

Therefore, our results underscore that artificial feeding presents significant challenges. It requires a greater number of blood feedings, leads to lower weight gain, and consequently extends the developmental time of the insects at each nymphal stage and throughout their life cycle.

Based on our findings, we conclude that feeding triatomines via artificial blood sources may limit or even prevent the successful sustainable rearing of these insects over time, thus hampering research on this group. Additionally, this method of blood offering may be inadvisable when introducing and establishing new colonies of insects captured in the field.

In such cases, the number of specimens is usually limited, and replicating bioclimatic conditions similar to their natural habitat in a laboratory setting is extremely challenging. Triatomines can be highly sensitive to variations in temperature and humidity [46–51]. Therefore, using artificial feeders for field-captured insects would introduce another complicating variable, further hindering the colonization of triatomines in an insectary environment.

We did not observe differences in survival rates or weight gain of the insects according to sex. In this regard, sex does not appear to influence the developmental dynamics of the two triatomine species studied. This result is consistent with expectations, as in triatomine biology, the only source of protein for all individuals comes from blood, regardless of sex.

## Effect of different blood offering methods on insect development

Parameters such as life cycle duration and mortality are critical variables that directly affect the population size of vectors and play a key role in the dynamics of rearing triatomine colonies in insectaries [52]. Colonies with excessively long developmental cycles can impede the execution of experiments and the distribution of samples to scientific collections, among other challenges.

A relevant aspect is that the life cycle of triatomines can vary based on species, environmental conditions, and the type of blood source used [20,53]. Literature indicated that *T. infestans* typically has a shorter developmental time than *P. megistus*, with both species primarily associated with birds and mammals in their natural environments [20;,54]. However, in our study, the average developmental time for *T. infestans*, regardless of the feeding method, was longer than that for *P. megistus*. Additionally, the average life cycle durations for both *T. infestans* and *P. megistus* were notably prolonged, as their developmental times exceeded those reported in earlier studies [20,22,25,38]. Our findings suggest that artificial

feeding adversely affects the developmental time of insects across most instars and in both species. Conversely, insects fed on live animal hosts displayed faster developmental times.

It has been noted that males generally emerge before females [20,55], often to meet the need for fertilizing females and ensuring species continuation or as a compensatory mechanism since males often have longer lifespans than females. In contrast, we did not observe significant differences in the total developmental time between females and males for either species when analyzing the different blood-offering methods as a whole.

## Association between treatments and insect survival

High mortality rates were observed across the different instars of *T. infestans* and *P. megistus* when fed via artificial feeders, with lower rates in groups where blood was provided through live animal hosts. Considerable mortality rates were also noted when insects of both species were fed alternately. Despite these observations, no statistically significant differences were found between the groups. This lack of significance is likely due to the presence of deaths among insects fed on live animal hosts, albeit at lower rates.

The high accumulation of deaths in triatomines fed via artificial feeders and those fed alternately throughout their development was so substantial that it surpassed the number of insects that reached adulthood (Fig 6). For *P. megistus*, the mortality rate in the group fed on live animal hosts was also notably high, nearly reaching half of the surviving insects (Fig 6). The source of the blood meal (chickens in this case) does not seem to be problematic, as birds are frequently used hosts for *P. megistus* in artificial environments, and these insects are often captured in chicken coops [54,56]. The high mortality observed in *P. megistus* may be related to their heightened sensitivity to handling, as they were weighed before and after feeding to monitor blood intake dynamics. Additionally, the high inbreeding within the colonies of both species studied, which have been maintained in the institutional insectary for over two decades without introducing new wild specimens, could be a contributing factor.

The highest mortality rates for both species occurred in the first two instars, although this mortality extended throughout the developmental cycle in artificial feeding. These findings align with previous studies, which reported mortality rates from 10 to 20% for the first instar in species of *Triatoma*, *Panstrongylus*, *Dipetalogaster*, and *Cavernicola* [57]. Specifically, *T. infestans* fed on chickens showed mortality rates of 11.5% during the N1 instar and 30.7% from N1 to adulthood[38]. In *P. megistus*, mortality rates of 16.7% for the entire cycle have been reported. Additionally, mortality rates as low as 4.8% were observed in three populations of *P. megistus* fed on mice and provided food twice a week [23].

Perlowagora-Szumlewicz (1953) [58] emphasized the importance of the duration of the first blood meal for successful molting. In our study, variations were minimized by exposing all first-instar nymphs of different species and experimental groups to feeding for the same period—30 minutes on the fifth day post-egg hatching. Studies have also reported that *T. infestans* N1 is ready to feed from the fourth day of life [22,59]. Factors such as the fragility of the mouthparts, difficulty in reaching the host, and capillary access in the first instar pose challenges for N1 to take their first meal. These factors are not present in artificial feeding, so this method should reduce physical difficulties for the insects. As such, lower mortality rates would be expected in the group fed artificially, but this was not observed in our trials. We suggest that the higher mortality of insects fed via artificial feeders might be due to an initial 'rejection' of the blood offered or the type of anticoagulant used. It is also important to note that *T. infestans* and *P. megistus* have a wide range of dietary sources [10] and have been reported to use rabbit blood as a food source in both natural and laboratory environments.

The anticoagulants used in the blood and their storage conditions affect the blood engorgement response of triatomines [60]. Therefore, the optimal anticoagulant for artificial feeding may depend on various factors, including insect species, anticoagulant concentration, and blood storage conditions [60].

While the literature provides numerous short-term trials using artificial feeding of triatomines [61–64]. Núnez & Lazzari (1990) [60] found that citrated bovine blood was more suitable than heparinized blood for feeding *T. infestans* nymphs, with citrate being an efficient blood preservative. According to these authors, the superiority of citrate is due to the stability

of cellular structures and phagostimulant molecules, while heparin has a weak blood preservation capacity, potentially producing deleterious compounds over time.

Buralli et al. (1980) [32] studied *T. infestans* populations fed on citrated, heparinized, defibrinated, and EDTA-added human blood. The authors achieved better results with citrated blood, with 85% imaginal molting. Niel (1980) [33] studied the behavior of *T. infestans* fed artificially on bird blood with citrate, fluoride, heparin, and oxalate anticoagulants, finding better results with heparinized blood. Ubiego et al. (1982) [65] compared the development of *T. infestans* fed on live chickens and citrated and defibrinated chicken blood, demonstrating that anticoagulants influenced species development, with live animal host feeding being superior to artificial feeding. The literature consistently shows that while *T. infestans* and *P. megistus* can feed on citrated blood, their development is constantly better with blood from live animal hosts.

### Number of feedings required for *P. megistus* molting in the three treatments

*Panstrongylus megistus* requires more contact events with the artificial feeder to reach the next developmental instar than insects fed on live animal hosts or alternately. This significant difference was observed for most molts except from N4 to N5.

Barbosa et al. (2001) [23] analyzed three *P. megistus* populations from different localities, feeding them on mice throughout their developmental cycle, with averages of 15, 16, and 18 feedings per population. In the present study, we observed that *P. megistus* fed on live animal hosts needed an average of 14.9 feedings to reach adulthood, while those fed alternately required 21.7 feedings, and the ones fed on artificial feeders needed 34.5 feedings. These findings reinforce that artificial feeding results in poorer developmental performance.

### Weight gain in different instars of *P. megistus*

*Triatoma infestans* and *Rhodnius prolixus* display the highest blood intake rates among triatomines [32]. Naturally, the amount of blood ingested influences the duration of each instar and the overall developmental cycle [38]. Blood intake rates are higher in the early nymphal instars, with insects able to ingest eight to nine times their body weight in blood [66]. Females ingest more blood than males, though both sexes ingest less at the N5 stage [20]. This greater blood investment by N5 is likely related to the development of anatomical structures and physiological changes during molting to adulthood [52,67]. At the adult stage, the volume of blood needed decreases, and insects increase their weight by two to four times. Schofield (1994) [66] argues that this process is related to the fact that adults have completed their development, and all ingested food is directed solely toward maintenance, survival, and reproductive activities.

Barbosa (1998) [68] studied three *P. megistus* populations and did not observe higher blood intake in females than in males. In the present study, females and males also ingested similar volumes of blood. However, this finding may be related to the small number of insects that reached adulthood in all experimental groups.

Overall, we found that the average weight gain of *P. megistus* fed on chickens increased throughout development and was significantly higher than that of artificially fed insects in most nymphal instars. However, in N1 and N5, no significant weight gain differences were observed between the groups. These findings correlate with the other results of the present study: insects ingesting larger volumes of blood complete the developmental cycle more quickly, as was the case for insects fed on live animal hosts.

Our results showed that despite operational ease, *in vitro* feeding for both *T. infestans* and *P. megistus* did not yield the expected success. Developmental time, weight gain, and fewer feeder contacts were more efficient in insects fed on live animal hosts (chickens) than in those fed artificially. For insects fed alternately, physiological parameters reached intermediate values and positions between the other two blood-offering methods.

Finally, we conclude this paper by emphasizing that, in experimental research involving animals, responsible scientists—anywhere in the world—do not wish to use animals or cause them unnecessary suffering if it can be avoided [40]. Furthermore, there are situations in which animals must continue to be used, because replacement practices and

resources prove difficult to accurately reproduce what is intended, such as in the case of imitating the complex physiological systems of entire living organisms, as in the research presented here. Additionally, the scientific community generally accepts the use of animals only within an ethical and protective framework. A series of independent surveys were conducted in the United Kingdom between 2002 and 2006 to understand society's perception of the use of animals in scientific experiments [69]. Interestingly, these surveys indicated that the general public accepts the use of animals in research for the benefit of human health. We concluded that the artificial feeding of triatomines, particularly the species studied does not represent a sustainable strategy for use in insectaries that require rapid production and a high number of triatomine specimens, and total feed replacement in vivo is not advisable. Furthermore, while anticoagulants may be suitable for laboratory experiments, *in vitro* feeding does not meet the objectives of full insect development. Finally, we conclude by highlighting that additional studies on other triatomine species are required to further explore this controversial topic.

## Supporting information

**S1 Text. GEE concept.**
(DOCX)

**S1 Table. Average developmental time (in days) of triatomines fed on chickens, artificial feeder, or alternating feeding methods across different instar stages.**
(PDF)

**S1 Fig. Number of insects reaching adulthood in *Triatoma infestans* according to sex.**
(TIF)

**S2 Fig. Number of insects reaching adulthood in *Panstrongylus megistus* according to sex.**
(TIF)

**S2 Table. Model of interaction between the different experimental groups and developmental instars when comparing the number of blood feedings required to trigger molting in *P. megistus*.** Interaction fixed at N1 and the artificial feeder group.
(PDF)

**S3 Table. Descriptive measures of weight gain by developmental instar of *P. megistus* across the three feeding methods.**
(PDF)

**S4 Table. Weight gain of N1–N5 nymphs in *P. megistus*.** Interaction between group and stage. The reference instar in all analyses was N1.
(PDF)

**S5 Table. Descriptive measures of weight gain according to sex for *P. megistus* fed on chickens, artificially and alternately.**
(PDF)

AcknowledmentsThe authors acknowledge the financial support of FAPEMIG, IRR and Coordination of health surveillance and reference laboratories (Fiocruz).

## Author contributions

**Conceptualization:** Liléia Gonçalves Diotaiuti, Silvia Ermelinda Barbosa.

**Data curation:** Raquel Aparecida Ferreira, Silvia Ermelinda Barbosa.

**Formal analysis:** Raquel Aparecida Ferreira, Eduardo Fernandes e Silva.

**Funding acquisition:** Liléia Gonçalves Diotaiuti, Silvia Ermelinda Barbosa.

**Investigation:** Ana Carolina Corrêa Nascimento, Natalya Lorena da Silva Santos, Silvia Ermelinda Barbosa.

**Methodology:** Silvia Ermelinda Barbosa.

**Project administration:** Silvia Ermelinda Barbosa.

**Resources:** Liléia Gonçalves Diotaiuti, Silvia Ermelinda Barbosa.

**Supervision:** Silvia Ermelinda Barbosa.

**Validation:** Raquel Aparecida Ferreira, Ana Carolina Corrêa Nascimento, Natalya Lorena da Silva Santos, Liléia Gonçalves Diotaiuti, Eduardo Fernandes e Silva, Silvia Ermelinda Barbosa.

**Visualization:** Raquel Aparecida Ferreira, Ana Carolina Corrêa Nascimento, Natalya Lorena da Silva Santos, Liléia Gonçalves Diotaiuti, Eduardo Fernandes e Silva, Silvia Ermelinda Barbosa.

**Writing – original draft:** Raquel Aparecida Ferreira, Silvia Ermelinda Barbosa.

**Writing – review & editing:** Raquel Aparecida Ferreira, Ana Carolina Corrêa Nascimento, Natalya Lorena da Silva Santos, Liléia Gonçalves Diotaiuti, Eduardo Fernandes e Silva, Silvia Ermelinda Barbosa.

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
