## [Decision Letter · Decision Letter 0]

9 Dec 2024

PONE-D-24-41986Impact of artificial feeding on the developmental cycle of two triatomine speciesPLOS ONE

Dear Dr. Ferreira,

Thank you for submitting your manuscript to PLOS ONE. After careful consideration, we feel that it has merit but does not fully meet PLOS ONE’s publication criteria as it currently stands. Therefore, we invite you to submit a revised version of the manuscript that addresses the points raised during the review process.

As there was some difficult in securing two reviews for your paper, I decided to go further with this single review. As you will see, the reviewer felt that the paper contributes to the field, but needs relevant changes pointed out in his review. One point that could help improve the paper's impact would be going further than just stating, as mentioned in the review, that in vitro is not the best way for triatomine rearing. In this sense, the review has several sound suggestions. Also, specific comments 4 and 6 are particularly critical, as methodological correctness is an essential for acceptance in PLos One.

We look forward to receiving your revised manuscript.

Kind regards,

Pedro L. Oliveira

Academic Editor

PLOS ONE

Journal Requirements:

2. Thank you for stating the following financial disclosure: [Financial support of Fundação de Amparo a Pesquisa de Minas Gerais (FAPEMIG), Instituto René Rachou (IRR) and Coordination of health surveillance and reference laboratories (Fiocruz).]. Please state what role the funders took in the study. If the funders had no role, please state: "The funders had no role in study design, data collection and analysis, decision to publish, or preparation of the manuscript." If this statement is not correct you must amend it as needed. Please include this amended Role of Funder statement in your cover letter; we will change the online submission form on your behalf.

3. Thank you for stating the following in your Competing Interests section: [NO authors have competing interests]. Please complete your Competing Interests on the online submission form to state any Competing Interests. If you have no competing interests, please state "The authors have declared that no competing interests exist.", as detailed online in our guide for authors at http://journals.plos.org/plosone/s/submit-now This information should be included in your cover letter; we will change the online submission form on your behalf.

4. We note that your Data Availability Statement is currently as follows: [All relevant data are within the manuscript and its Supporting Information files.] Please confirm at this time whether or not your submission contains all raw data required to replicate the results of your study. Authors must share the “minimal data set” for their submission. PLOS defines the minimal data set to consist of the data required to replicate all study findings reported in the article, as well as related metadata and methods (https://journals.plos.org/plosone/s/data-availability#loc-minimal-data-set-definition). For example, authors should submit the following data: - The values behind the means, standard deviations and other measures reported; - The values used to build graphs; - The points extracted from images for analysis. Authors do not need to submit their entire data set if only a portion of the data was used in the reported study. If your submission does not contain these data, please either upload them as Supporting Information files or deposit them to a stable, public repository and provide us with the relevant URLs, DOIs, or accession numbers. For a list of recommended repositories, please see https://journals.plos.org/plosone/s/recommended-repositories. If there are ethical or legal restrictions on sharing a de-identified data set, please explain them in detail (e.g., data contain potentially sensitive information, data are owned by a third-party organization, etc.) and who has imposed them (e.g., an ethics committee). Please also provide contact information for a data access committee, ethics committee, or other institutional body to which data requests may be sent. If data are owned by a third party, please indicate how others may request data access.

Reviewers' comments:

Reviewer's Responses to Questions

**Comments to the Author**

1. Is the manuscript technically sound, and do the data support the conclusions?

Reviewer #1: Partly

2. Has the statistical analysis been performed appropriately and rigorously? 

Reviewer #1: Yes

3. Have the authors made all data underlying the findings in their manuscript fully available?

Reviewer #1: Yes

4. Is the manuscript presented in an intelligible fashion and written in standard English?

Reviewer #1: No

5. Review Comments to the Author

Reviewer #1: General comment

The manuscript submitted by Ferreira and co-workers presents a detailed analysis of the effect of three feeding conditions on different parameters in two species of triatomine bugs with public health relevance as vectors of Chagas disease.

Mass-rearing of triatomine bugs is a key activity in basic and applied research on triatomine biology, parasite studies and infection diagnosis. Optimising the effort and cost of such a demanding task is essential in order to supply large numbers of insects on a regular basis.

In vitro feeding is a technique that has been used for several decades to rear haematophagous arthropods, with varying degrees of success depending on the species.

In the case of triatomines, much work has been done to find the most appropriate conditions and the method has been successfully adopted for rearing small to medium sized laboratory populations. Large-scale rearing as required, for example in reference centres or diagnostic units, usually involves feeding bugs on live hosts, which requires specific infrastructure and personnel. Modern ethical guidelines on animal care and welfare require the search for alternatives to feeding insects on live hosts.

The aim of the present manuscript is to provide a comparison of biological parameters that could be relevant for mass rearing, in bugs reared in vivo, in vitro and alternating both conditions.

Even though, the study has been conducted at a small scale, the data collected could be of interest; yet, some points deserve further attention by the authors.

The aim of the present manuscript is providing a comparison on biological parameters, in bugs reared in vivo, in vitro and alternating both conditions.

Specific points

1) Line 56 “Vinchuca” is not the only vernacular Spanish name given to triatomines. “Chinche besucona”, “chinche”, “pito” and also employed, depending on the country. Please, check and complete.

2) Lines 60-61. The feasibility of rearing triatomines in vitro has long been demonstrated. Please rephrase to make it clear that you are interested in large-scale rearing.

3) Lines 68-70. Please rephrase, it seems impractical for your needs and under your conditions, but it has been used successfully for decades.

4) Lines 169-175. As indicated in several cited references, it is crucial to precise the treatment and conditioning of the blood used for in vitro feeding. Parameters as how long after extraction blood was proposed to bugs, how was it stored in between, the exact composition and amount of anticoagulant solutions, the exact nature and treatment of latex membranes, the type of artificial feeder used, the exact temperature of the blood (and not that of the water bath) are just some of them.

5) The authors repeatedly point out that previous reports on in vitro feeding of triatomines do not follow the bugs over several generations, as an argument to justify the need for their study. However, the manuscript only presents what happens along a single generation, so the previous criticism applies to them. My advice would be to provide solid arguments justifying the rationale for a study in 2024 that is not much different from many others done decades ago.

6) A major inconsistency that seems unjustified in the current version is why the authors chose to use different species as blood donors and for live host feeding. Why not use the same blood? This would have revealed differences in the methods, whereas in the present experiment the effect of donor and method cannot be distinguished. This reason is probably practical, but needs to be explained.

7) Some sections of the manuscript are inappropriately long. The detailed description of the statistical treatment of the data is interesting, but can be presented in supplementary material. The results section is also unnecessarily long, but for a different reason: there is redundancy in the information presented in the tables and in the text.

8) The discussion focuses on the advantages of in vivo over in vitro feeding, which, as the authors say in the introduction, is now well acknowledged. Simply stating that in vitro is not the best and that this strategy might be potentially problematic for mass rearing is much less than most educated readers of PLoS One would expect. In the opinion of this reviewer, this study provides an interesting opportunity to discuss the trade-off that scientists face today, i.e. optimising efforts and resources to advance public health science versus animal welfare regulations. In particular in Chagas endemic countries, where the need of advances is high and the resources usually limited. This problematic goes beyond Brazilian legislation and triatomine rearing.

6. PLOS authors have the option to publish the peer review history of their article (what does this mean? ). If published, this will include your full peer review and any attached files.

**Do you want your identity to be public for this peer review?** For information about this choice, including consent withdrawal, please see our Privacy Policy .

Reviewer #1: No

---

## [Author Response · Author response to Decision Letter 1]

7 Mar 2025

Dear reviewers, we appreciate your suggestions and have accepted almost all of them. Below are the point-by-point responses.

1) Line 56 “Vinchuca” is not the only vernacular Spanish name given to triatomines. “Chinche besucona”, “chinche”, “pito” and also employed, depending on the country. Please, check and complete.

Authors: The names were added as suggested by the reviewer

2) Lines 60-61. The feasibility of rearing triatomines in vitro has long been demonstrated. Please rephrase to make it clear that you are interested in large-scale rearing.

Authors: Changes were made as suggested.

3) Lines 68-70. Please rephrase, it seems impractical for your needs and under your conditions, but it has been used successfully for decades.

Authors: Changes were made as suggested.

4) Lines 169-175. As indicated in several cited references, it is crucial to precise the treatment and conditioning of the blood used for in vitro feeding. Parameters as how long after extraction blood was proposed to bugs, how was it stored in between, the exact composition and amount of anticoagulant solutions, the exact nature and treatment of latex membranes, the type of artificial feeder used, the exact temperature of the blood (and not that of the water bath) are just some of them.

Authors: Thank you for your comments. Please note that the information has been added in lines 206-221. The blood used in the artificial feedings was 10% citrated rabbit blood (25g P.A. citrate, 8g citric acid, and 24.5g glucose). This blood is collected, prepared, and sterilized at the Institute of Science and Technology in Biomodels (ICTB)/Fiocruz. It is packaged in an autoclavable glass bottle with a rubber cap, sealed, and distributed chilled on dry ice to the laboratories. In the laboratory, the blood is immediately stored at -20°C and used within 10 days of being collected.

The artificial feeding device used consisted of a glass apparatus, made by a craftsman in the shape of a baby bottle, with an external compartment for circulating water heated by an attached water bath and an internal compartment containing the blood. The water temperature was set at 40°C, and the blood temperature in the bottles remained at 39.5°C. The membrane used to cover the feeding bottle was a latex membrane. These were obtained from natural latex gloves, which were cut into 10x10 cm pieces and washed with Type I water to remove dust. The dried latex membrane was used to seal the mouth of the glass apparatus with the help of an elastic band, simulating the skin of a live animal that was pierced by the insect's mouthparts during feeding (Lazzari, 2014).

5) The authors repeatedly point out that previous reports on in vitro feeding of triatomines do not follow the bugs over several generations, as an argument to justify the need for their study. However, the manuscript only presents what happens along a single generation, so the previous criticism applies to them. My advice would be to provide solid arguments justifying the rationale for a study in 2024 that is not much different from many others done decades ago.

Authors: The passage between lines 110 and 113 was moved to lines 131–135. The sentence, 'One of these gaps involves the effectiveness and suitability of rearing insects exclusively with artificial feeders over multiple generations in an insectary,' between lines 115 and 118, was removed from the manuscript. Additionally, the aspect highlighted by the reviewer was revised throughout the manuscript (lines 573–580; 587–591; 594-598;764–776), and minor changes were made to the writing of the last paragraph of the introduction. We hope these adjustments have met your expectations.

6) A major inconsistency that seems unjustified in the current version is why the authors chose to use different species as blood donors and for live host feeding. Why not use the same blood? This would have revealed differences in the methods, whereas in the present experiment the effect of donor and method cannot be distinguished. This reason is probably practical, but needs to be explained.

Authors: We understand the reviewer's questions and hope to clarify this point. It is important to note that working with live animals has not been an easy task in the scientific environment, particularly in Brazil. This work is the result of various questions and recommendations from the Ethics Committee on the Use of Animals. The license obtained from CEUA to keep triatomines in an insectary, after a 2.5-year process, does not allow us to breed rabbits at our institution to be used as a source of blood for the triatomines. For this reason, we use the license granted to ICTB/Fiocruz (as mentioned in question 4). ICTB is the technical-scientific unit of Fiocruz responsible for studies, development, and production in Laboratory Animal Science. They prepare and supply blood from other animals, such as rodents and lagomorphs, to other Fiocruz units. Although we have a license to use chickens to provide blood for feeding triatomines kept in the institutional insectary, we lack the expertise and equipment at our institution to prepare the blood, monitor the health of the animals, and carry out pre-clinical analyses on animals bred to ensure quality control of the blood for use in the artificial feeder. Therefore, for practical reasons, the same blood donors were not used in the tests in this study.

7) Some sections of the manuscript are inappropriately long. The detailed description of the statistical treatment of the data is interesting, but can be presented in supplementary material. The results section is also unnecessarily long, but for a different reason: there is redundancy in the information presented in the tables and in the text.

Authors: We appreciate your suggestions. Regarding the statistical analyses, we have removed the paragraph describing the GEE test from the manuscript. However, due to the complexity of the analyses and models employed, we believe that the details and description of the criteria used in decision-making should remain in the manuscript.

Regarding the results section, we completely agree with the reviewer and have removed the redundant information.

8) The discussion focuses on the advantages of in vivo over in vitro feeding, which, as the authors say in the introduction, is now well acknowledged. Simply stating that in vitro is not the best and that this strategy might be potentially problematic for mass rearing is much less than most educated readers of PLoS One would expect. In the opinion of this reviewer, this study provides an interesting opportunity to discuss the trade-off that scientists face today, i.e. optimising efforts and resources to advance public health science versus animal welfare regulations. In particular in Chagas endemic countries, where the need of advances is high and the resources usually limited. This problematic goes beyond Brazilian legislation and triatomine rearing.

Authors: We appreciate your comments. We have expanded the discussion, addressing the points highlighted by the reviewer. A paragraph was added between lines 573 and 580, and a sentence was added between lines 587-591 and 594-598. Finally, in the last paragraph, lines 774-776 were added, and the ending of this paragraph was rewritten. We hope that these additions are sufficient, as we do not intend to engage in discussions that involve, including, philosophical aspects, which would be more appropriate for another type or format of manuscript.

---

## [Decision Letter · Decision Letter 1]

2 Apr 2025

Impact of artificial feeding on the developmental cycle of two triatomine species

PONE-D-24-41986R1

Dear Dr. Ferreira,

We’re pleased to inform you that your manuscript has been judged scientifically suitable for publication and will be formally accepted for publication once it meets all outstanding technical requirements.

Kind regards,

Pedro L. Oliveira

Academic Editor

PLOS ONE

Additional Editor Comments (optional):

Reviewers' comments:

Reviewer's Responses to Questions

**Comments to the Author**

1. If the authors have adequately addressed your comments raised in a previous round of review and you feel that this manuscript is now acceptable for publication, you may indicate that here to bypass the “Comments to the Author” section, enter your conflict of interest statement in the “Confidential to Editor” section, and submit your "Accept" recommendation.

Reviewer #1: All comments have been addressed

2. Is the manuscript technically sound, and do the data support the conclusions?

Reviewer #1: Yes

3. Has the statistical analysis been performed appropriately and rigorously? 

Reviewer #1: Yes

4. Have the authors made all data underlying the findings in their manuscript fully available?

Reviewer #1: Yes

5. Is the manuscript presented in an intelligible fashion and written in standard English?

Reviewer #1: Yes

6. Review Comments to the Author

Reviewer #1: Thank you for having adequately addressed my comments. In the opinion of this reviewer, the manuscript has been improved in the new version.

7. PLOS authors have the option to publish the peer review history of their article (what does this mean? ). If published, this will include your full peer review and any attached files.

**Do you want your identity to be public for this peer review?** For information about this choice, including consent withdrawal, please see our Privacy Policy .

Reviewer #1: No

---

## [Editor Report · Acceptance letter]

PONE-D-24-41986R1

PLOS ONE

Dear Dr. Ferreira,

I'm pleased to inform you that your manuscript has been deemed suitable for publication in PLOS ONE. Congratulations! Your manuscript is now being handed over to our production team.

Kind regards,

on behalf of

Dr. Pedro L. Oliveira

Academic Editor

PLOS ONE